# Effects of Psychotropic Medication on Somatic Sterol Biosynthesis of Adult Mice

**DOI:** 10.3390/biom12101535

**Published:** 2022-10-21

**Authors:** Marta Balog, Allison C Anderson, Marija Heffer, Zeljka Korade, Karoly Mirnics

**Affiliations:** 1Munroe-Meyer Institute for Genetics and Rehabilitation, University of Nebraska Medical Center, Omaha, NE 68105, USA; 2Department of Medical Biology and Genetics, Faculty of Medicine, J. J. Strossmayer University of Osijek, 31000 Osijek, Croatia; 3Department of Pediatrics, College of Medicine, University of Nebraska Medical Center, Omaha, NE 68198, USA; 4Department of Biochemistry and Molecular Biology, College of Medicine, University of Nebraska Medical Center, Omaha, NE 68198, USA; 5Child Health Research Institute, University of Nebraska Medical Center, Omaha, NE 68198, USA; 6Department of Psychiatry, College of Medicine, University of Nebraska Medical Center, Omaha, NE 68198, USA; 7Department of Pharmacology and Experimental Neuroscience, College of Medicine, University of Nebraska Medical Center, Omaha, NE 68198, USA

**Keywords:** aripiprazole, trazodone, polypharmacy, 7-dehydrocholesterol, 7-dehydrocholesterol reductase, cholesterol, desmosterol, sterols

## Abstract

Polypharmacy is commonly used to treat psychiatric disorders. These combinations often include drugs with sterol biosynthesis inhibiting side effects, including the antipsychotic aripiprazole (ARI), and antidepressant trazodone (TRZ). As the effects of psychotropic medications are poorly understood across the various tissue types to date, we investigated the effects of ARI, TRZ, and ARI + TRZ polypharmacy on the post-lanosterol biosynthesis in three cell lines (Neuro2a, HepG2, and human dermal fibroblasts) and seven peripheral tissues of an adult mouse model. We found that both ARI and TRZ strongly interfere with the function of 7-dehydrocholesterol reductase enzyme (DHCR7) and lead to robust elevation in 7-dehydrocholesterol levels (7-DHC) and reduction in desmosterol (DES) across all cell lines and somatic tissues. ARI + TRZ co-administration resulted in summative or synergistic effects across the utilized in vitro and in vivo models. These findings suggest that at least some of the side effects of ARI and TRZ are not receptor mediated but arise from inhibiting DHCR7 enzyme activity. We propose that interference with sterol biosynthesis, particularly in the case of simultaneous utilization of medications with such side effects, can potentially interfere with functioning or development of multiple organ systems, warranting further investigation.

## 1. Introduction

Physicians treating patients with schizophrenia, bipolar disorder or major depression frequently encounter elaborate clinical cases that have to be managed by complex medication regimes. Some of these situations require psychotropic polypharmacy treatment [1,2]. Unfortunately, while polypharmacy can greatly benefit patients, it can also result in a higher susceptibility for development of metabolic syndrome [1] or other adverse effects [3,4,5,6,7,8,9].

Many of the psychotropic polypharmacy combinations include drugs with sterol biosynthesis inhibiting side effects. One of the commonly used combinations of psychotropic medications is simultaneous utilization of the antipsychotic aripiprazole (ARI), and antidepressant trazodone (TRZ) [10,11,12]. TRZ selectively inhibits neuronal serotonin reuptake and is an antagonist of 5-HT2A receptors. In addition, TRZ shows antagonism at 5-HT2B, 5-HT2C, adrenergic alpha-1, and partial agonism at 5-HT1A receptors [13,14]. In contrast, ARI is a quinolinone antipsychotic that is a partial agonist at the D2 and 5HT1aA receptors and an antagonist at the 5HT-2aA receptor. It has a high affinity for D2, D3, 5-HT-1aA, and 5-HT2aA receptors and moderate affinity for D4, 5-HT2C, 5-HT7, alpha-1 adrenergic, and H1 receptors [15].

ARI and TRZ have been also shown to increase 7-DHC in the CNS in various developmental in vitro and in vivo models, as well as human biomaterials [16,17,18,19,20] but ARI and TRZ treatment effects in adult peripheral organs remain unknown. Sterol biosynthesis inhibition by ARI and TRZ is mediated through blocking the effect of 7-dehydrocholesterol reductase (DHCR7) enzyme, which simultaneously blocks the conversion of 7-dehydrocholesterol (7-DHC) to cholesterol, and conversion of 7-dehydrodesmosterol (7-DHD) to desmosterol (DES) [21]. Ultimately, this results in two main effects of this DHCR7 inhibition—a sharp increase in 7-DHC levels and a substantial decrease in DES levels. Notably, 7-DHC is the most oxidizable lipid known to date, with a propagation rate constant of 2160 (this is 200 times more than cholesterol and 10 times more than arachidonic acid) [22,23]. The result of spontaneous 7-DHC peroxidation is formation of highly reactive autoxidation sterols, called 7-DHC derived oxysterols [24,25]. 7-DHC derived oxysterols have multiple bioactive effects, and these reactive electrophiles can affect cell viability, differentiation, and growth [25,26,27].

Cholesterol biosynthesis occurs in all type of cells and is essential for cellular homeostasis and structural integrity [28]. As the effects of ARI and TRZ utilizations are poorly understood across the various tissue types to date, we investigated the effects of ARI, TRZ and ARI + TRZ polypharmacy on the post-lanosterol peripheral sterol biosynthesis in three cell lines (Neuro2a, HepG2, and human dermal fibroblasts) and an adult mouse model. Study design is presented in Figure 1.

## 2. Materials and Methods

### 2.1. Reagents

Majority of reagents used were purchased from Sigma-Aldrich Co (St. Louis, MO, USA) unless noted differently. Solvents used for HPLC were purchased from ThermoFisher Scientific Inc. (Waltham, MA, USA). ARI and TRZ were obtained from Selleckchem (Radnor, PA, US) and were prepared in sterile DMSO solution. All sterol standards used, natural and isotopically labeled, are available from Kerafast, Inc. (Boston, MA, USA). 

### 2.2. Aripiprazole and Trazodone In Vivo Treatment

Three months old male mice, C57Bl/6J (stock number: 000664), were purchased from The Jackson Laboratories. Mice were housed at 12 h light/dark cycle, at a constant temperature (25 °C) and humidity (40–60%) in standard ventilated mouse cages with ad libitum access to food (Teklad LM-485 Mouse/Rat Sterilizable Diet 7012) and water in Comparative Medicine at the University of Nebraska Medical Center (UNMC), Omaha, NE, USA. In humans, TRZ (Desyrel) is given at a starting dose 150 mg/day and may be increased by 50 mg per day to a maximum dose of 400 mg per day. TRZ is often prescribed as sleep aid at a starting dose of 50 mg/day. In case of a dose of 50 mg/60 kg human body mass, this calculation leads to a dose of 0.83 mg/kg/day. Animal Equivalent dose (AED in mg/kg) is calculated as human dose (mg/kg) 50 mg per day) × Km ratio (12.3) = 10 mg/kg [29]. We used a low dose of 10 mg/kg for the experimental treatment of mice. We applied the same calculations and literature data for the second drug, ARI (Abilify) and decided to use it at 2.5 mg/kg for treatment in mice (this corresponds to ARI dose of 10–15 mg/day in humans). Most commonly used doses of ARI in humans are 2 mg–30 mg/day [30]. We used 35 mice in our study with 9 animals assigned to all groups, except for control group that consisted of 8 mice. Intraperitoneal injections were used for drug (or vehicle) delivery, every day at 8.00 am. The treatment did not influence body mass of animals during experiment. All procedures were performed in accordance with the Guide for the Humane Use and Care of Laboratory Animals. The use of mice in this study was approved by the Institutional Animal Care and Use Committee of UNMC.

### 2.3. Tissue Dissection and Preparation for Sterol Analysis

Mice were anesthetized with Isoflurane overdose (Forane^®^ isofluranum, Abbott Laboratories LTD; Lake Bluff, IL, USA) four to six hours after the last treatment was applied. Seven organs were dissected: heart, lung, kidney, liver, spleen, pancreas, and adrenal gland. Heart and serum were evaluated because of cardiovascular role, spleen because of the role in immunity, liver, pancreas, and kidney because of their involvement in metabolic syndrome, and adrenal gland because of stress response and steroid hormone synthesis. All tissues were frozen in pre-chilled methyl-butane and stored at −80 °C. Blood was collected and centrifuged for serum collection, all samples were frozen and stored at −80 °C.

Frozen samples were sonicated in pre-chilled PBS buffer containing butylated hydroxytoluene (BHT) and triphenylphosphine (PPh3). For sterol measurements (first set of aliquots) we used 10 µL which corresponds to approximately150–250 µg of protein. For protein measurements (second set of aliquots) we used 20 µL of the same sample solution, diluted it 10 and 20 times and measured protein concentration in these diluted samples. For drug level measurements (third set of aliquots) we used 100 µL of starting sample which corresponds to approximately 1.5–2.5 milligrams of tissue. The protein was measured using BCA assay (Pierce). Sterol levels were normalized to protein measurements and expressed as nmol/mg protein. The third set of aliquots of homogenized tissue were used for drug measurements. Sterol levels in serum were expressed as nmol/mL.

### 2.4. LC-MS/MS Sterol Measurements

Sterols were extracted and derivatized with PTAD as described previously [31] and placed in an Acquity UPLC system equipped with ANSI-compliant well plate holder coupled to a Thermo Scientific TSQ Quantis mass spectrometer equipped with an APCI source. After that, 10 μL of sample was injected onto the column (Phenomenex Luna Omega C18, 1.6 μm, 100 Å, 2.1 mm × 100 mm) with 90% MeOH and 10% ACN (0.1% *v*/*v* acetic acid) mobile phase for 1.7 min runtime at a flow rate of 500 μL/min. Natural sterols were analyzed by selective reaction monitoring (SRM) using the following transitions: Chol 369 → 369, 7-DHC 560 → 365, DES 592 → 560, lanosterol 634 → 602, with retention times of 0.7, 0.4, 0.3 and 0.3 min, respectively. SRMs for the internal standards were set to: d_7_-Chol 376 → 376, d_7_-7-DHC 567 → 372, ^13^C_3_-DES 595 → 563, ^13^C_3_-lanosterol 637 → 605. 

### 2.5. ARI and TRZ Measurements 

Drugs were extracted from all sample types using methyl tert-butyl ether and ammonium hydroxide as described previously [32]. ARI levels were acquired in an Acquity UPLC system coupled to a Thermo Scientific TSQ Quantis mass spectrometer using an ESI source in the positive ion mode. Ten μL of each sample was injected onto the column (Phenomenex Luna Omega C18, 1.6 μm, 100 Å, 2.1 × 50 mm2) using water (0.1% *v*/*v* acetic acid) (solvent A) and acetonitrile (0.1% *v*/*v* acetic acid) (solvent B) as mobile phase. The gradient was: 10–40% B for 0.5 min; 40–95% B for 0.4 min; 95% B for 1.5 min; 95–10% B for 0.1 min; 10% B for 0.5 min. ARI, TRZ and their metabolites were analyzed by SRM using the following transitions: ARI 448 → 285, dehydroaripiprazole 446 → 285, TRZ 372 → 176, *m*-CPP 197 → 153. The SRM for the internal standards (d8-ARI and d8-m-CPP) were set to 456 → 293 and 205 → 157, respectively. Final medications levels are reported as ng/mg of protein for analyzed organs and ng/ml for serum samples. 

### 2.6. Cell Cultures 

Human hepatocellular carcinoma HepG2 cells and mouse neuroblastoma Neuro2a cells were purchased from ATCC (Rockville, MD, USA). Control human fibroblasts were obtained from Coriell Institute for Medical Research (Camden, NJ, USA). HepG2 and human dermal fibroblasts were maintained in DMEM with 10% fetal bovine serum and Neuro2a were maintained in EMEM with 10% fetal bovine serum. To determine the effect of drugs, cells were plated in 96-well plates and incubated at 37 °C in 5% CO_2_ for 48 h in presence and absence of different concentrations of ARI, TRZ, and ARI + TRZ. The treatment was performed in cholesterol deficient medium. HepG2 and human fibroblast cultures were grown in DMEM with 10% delipidated fetal bovine serum, and Neuro2a were grown in EMEM plus N2 supplement. At the end point of the incubation, Hoechst dye was added to all wells in the 96-well plate, and the total number of cells was counted using an ImageXpress Pico and cell counting algorithm using CellReporterXpress. After removing the medium, wells were rinsed twice with 1× PBS and then stored at −80 °C for sterol analysis. All samples were analyzed within 2 weeks of freezing.

### 2.7. Statistical Analyses 

Statistical analyses were performed using GraphPad Prism 9 for Windows. Unpaired two-tailed t-test was applied for individual comparisons between two groups. Discovery was determined using the two-stage linear step-up procedure of Benjamini, Krieger, and Yekutieli, with Q = 5%. Each analysis was performed individually, without assuming a consistent SD. Statistical test showed a normal distribution in all comparison groups. The p-values for statistically significant differences are highlighted in the figures and/or figure legends.

## 3. Results

### 3.1. ARI + TRZ Polypharmacy Increases 7-DHC Levels in HepG2, Neuro2a and Human Fibroblast Cell Cultures Compared to Single Drug Treatment

HepG2, Neuro2a, and human fibroblast cells were treated with ARI, TRZ, or ARI + TRZ. HepG2 and Neuro2a cells were treated for 48h: HepG2 with five different concentrations of ARI, TRZ, or ARI + TRZ and Neuro2a with six different concentrations. Control human fibroblasts were treated with five different concentrations of ARI, TRZ, or ARI + TRZ for 7 days. CHOL, DES, 7-DHC, and LAN were analyzed with PTAD derivatization assay by LC-MS/MS. LC-MS/MS analysis revealed that treatment did not affect cell viability, however sterol profile was significantly altered by the treatment. Figure 2 depicts 7-DHC and DES levels, as well as 7-DHC/CHOL ratio for treatment with ARI, TRZ, or ARI + TRZ in comparison to vehicle treated controls. The response of 7-DHC, DESM, CHOL, and 7-DHC fold change to all concentrations of ARI, TRZ, and ARI + TRZ drug treatments can be found in Appendix A. 

In our experiments, we refer to the combined effect of medication as *summation* when the combination treatment (ARI + TRZ) exceeds the effects of the individual medications (ARI or TRZ). *Synergy* is defined as a change by combination treatment (ARI + TRZ) that exceeds the mathematical sum of the two individual treatments (ARI and TRZ). Note that treatment with ARI + TRZ had a synergistic effect on 7-DHC increase in HepG2 and Neuro2a cells (Figure 2A,B, denoted as fold-change of 7-DHC/vehicle treated in Appendix A) and showed summation in human fibroblast cultures (Figure 2C and Appendix A). 

DES levels (Figure 2A–C, middle panel) were most profoundly decreased in Neuro2a cells compared to control treatment, with ARI + TRZ having the strongest effect. In HepG2 cells, DES was significantly decreased compared to VEH-treated controls, with all three treatments showing a comparable magnitude of DES change. DES was the least changed in human fibroblasts, with only TRZ treatment reaching significantly decreased effect. Full DES profile is presented in Appendix A.

7-DHC/CHOL ratio (Figure 2A–C, right panel) was increased by 19-fold over vehicle-treated controls upon ARI treatment (50 nM), 38-fold by TRZ treatment (250 nM), and 70-fold by co-administration of ARI + TRZ (50 + 250 nM) in HepG2 cells. In Neuro2a 7-DHC/CHOL ratio was increased by 5-fold by ARI (25 nM), 9-fold by TRZ (25 nM), and 50-fold by co-administration of ARI + TRZ (25 + 25 nM). In both HepG2 and Neuro2a cells 7-DHC/CHOL ratio reached synergistic effect with ARI + TRZ co-administration. In human fibroblasts combined treatment reached summation: ARI treatment (50 nM) caused 25-fold increase in 7-DHC/CHOL ratio, TRZ (250 nM) an increase of 57-fold, and ARI + TRZ 81-fold. 

The observed effects were dose dependent to a large degree. The dose responses for all tested drug concentrations are presented in Appendix A. We found that 7-DHC was increased the most in Neuro2a cultures up to 250-fold for ARI (500 nM), 220-fold for TRZ (500 nM), and 284-fold for ARI+TRZ (500 + 500 nM) treatment compared to vehicle treated controls. In Neuro2a cells, synergy in 7-DHC increase was present up to 100 nM concentration of both drugs and their combined treatment. In HepG2 cells synergistic effect of 7-DHC increase was present even at the highest concentrations of drug treatments – it reached 5-fold increase for ARI (100 nM), 74-fold increase for TRZ (500 nM), and 81-fold increase for ARI+TRZ (100 + 500 nM). 7-DHC increase in human fibroblasts reached summation in all used concentrations and was 42-fold increased for ARI (100 nM), 62-fold increased for TRZ (500 nM), and 77-fold increased for ARI+TRZ (100 + 500 nM).

Significant reductions in CHOL were observed only in treated human fibroblasts and this was similar across all three treatments and all concentrations used (Appendix A). The low turnover rate of cholesterol [33,34] and a relatively short period of treatment under in vitro conditions might explain that a similar effect on CHOL was not observed in HepG2 and Neuro 2A cells. LAN concentrations were not affected by either treatment used. These data suggest that under in vitro conditions the effects of ARI + TRZ polypharmacy are synergistic or summative, and preferentially affect 7-DHC levels.

### 3.2. Baseline Sterols Levels Differ across Peripheral Tissues of Adult Mice

Baseline free sterol levels showed considerable differences across the serum and seven analyzed organs of sham-treated adult mice (Figure 3A). We observed the highest concentration of free CHOL and its two immediate precursors, 7-DHC and DES, in the serum samples. These three tested analytes showed somewhat different distribution across the organs: kidneys had the highest basal 7-DHC levels, spleen and lungs showed the highest DES concentration, and lung revealed the highest free CHOL content. The lowest basal 7-DHC and DES levels were observed in pancreas samples, while heart had the lowest free CHOL. 7-DHC/CHOL and DES/CHOL ratios also confirmed the variability across the investigated organs, with heart and adrenal gland having the highest DESM/CHOL ratio and kidney having the highest 7-DHC/CHOL ratio (Figure 3B). This suggests various sterol requirements and homeostasis in different peripheral organs, potentially related to different rate of biosynthesis and/or turnover. Notably, our measurements focused on free cholesterol only, and the observed relationships are potentially different for esterified cholesterol content [35,36].

### 3.3. ARI, TRZ, and Their Metabolites Are Detectable in the Serum and Peripheral Organs of Treated Mice

Experimental mice were treated with intraperitoneal injections of either VEH, 2.5 mg/kg ARI, 10 mg/kg TRZ or combined medications (ARI + TRZ). Mice were injected daily for 8 days. To confirm that the drugs reached the peripheral tissues, ARI and TRZ concentrations were measured in serum and seven peripheral organs in the ARI + TRZ group (Figure 4). ARI and its active metabolite—dehydro-ARI—were detected in the serum and in all seven examined peripheral organs. Dehydro-ARI was found in approximately 3–4-fold lower concentrations than ARI in all samples. TRZ was detected in all peripheral tissues, as well as its active metabolite *meta*-chlorophenylpiperazine (*m*-CPP), with very similar concentrations in all sample types. ARI and its metabolite (dehydro-ARI) reached the highest concentrations in serum and pancreas, while TRZ and its metabolite (*m*-CPP) were less variable across different peripheral organs. Notably, serum contained the highest absolute amounts of both medications and their metabolites. Corresponding data were obtained in mice treated with a single drug (TRZ or ARI) (data not shown). No medications or metabolites were detected in the vehicle-treated animals.

### 3.4. ARI and TRZ Increased 7-DHC in a Synergistic Fashion in Serum, Liver, and Spleen and Decreased DES in All Sample Types

CHOL, 7-DHC, DES, and LAN levels were assessed in the serum and seven peripheral organs of vehicle- and drug-treated mice. When compared to VEH-group, both ARI and TRZ animal groups had significantly increased 7-DHC concentrations. Moreover, ARI + TRZ treatment caused significant synergistic 7-DHC increase in serum, liver, and spleen samples, when compared to single ARI and TRZ effects alone (Figure 5). In the other tissues, the combined treatment of ARI + TRZ resulted in summation. Moreover, 7-DHC/CHOL ratio was increased upon all treatments (Appendix A), with the highest increase observed in serum, liver, and spleen.

DES levels were significantly decreased in serum and all examined organs upon single and combined treatments compared to VEH-treated group (Figure 6). In peripheral organs, ARI increased 7-DHC concentrations 4–16-fold, TRZ caused 5–10-fold increase while combined ARI + TRZ treatment caused 7–38-fold increase. DES decrease mirrored 7-DHC changes, with ARI + TRZ treatment causing the most prominent decrease in serum and examined organs. ARI caused average overall DES to decrease by 39%, TRZ by 46% and ARI + TRZ by 52% across serum and peripheral organs (Appendix A, lower panel).

Mean concentrations of 7-DHC and DES in all groups for serum and seven analyzed peripheral organs are presented in Table 1**.** Serum and all organs were affected by all three different treatments, however combined ARI + TRZ treatment caused synergistic 7-DHC increase in spleen (38-fold increase), serum (34-fold increase), and liver (25-fold increase) (Appendix A, upper panel). Strongest effect on DES was observed in serum (56% decrease) and spleen (52% decrease), while the least affected was pancreas (16% decrease) (Appendix A, lower panel). LAN or CHOL levels were unchanged (data not shown). Compared to cell cultures which are actively dividing, cholesterol in organs did not change because cholesterol synthesis is very tightly controlled due to its highly important metabolic and structural role [25,26]. 

## 4. Discussion

Psychiatric therapies often encompass simultaneous use of multiple psychotropic medications due to complexity of symptoms, unresponsiveness, or symptom resistance of patients to treatment [2]. Up to 30% of psychiatric patients are treated with two or more psychotropic medications [37,38], with many side-effects reported. Unfortunately, the majority of these studies were not able the address the underlying molecular mechanisms resulting in the adverse effects. 

The side-effects of antipsychotic polypharmacy treatments have been mostly attributed to the molecular mechanisms related to their main effect, which is achieved by modulating the activity of membrane-embedded proteins in the brain (e.g., receptors and transporters) [39]. Our study offers an alternative explanation for at least a subset of the side effects of the ARI-TRZ polypharmacy: their sterol biosynthesis modulating effect through inhibition of the DHCR7 enzyme activity. The DCHR7 inhibition leads to sharp rise in 7-DHC, which undergoes spontaneous peroxidation and gives rise to 7-DHC derived oxysterols, such as DHCEO. These 7-DHC derived oxysterols are biologically active compounds and can affect cell viability and growth [24,40,41]. 

Notably, DHCR7 inhibition by ARI and TRZ affects all tissues of the body, and 7-DHC elevation appears to be a hallmark in every organ we investigated. Thus, ARI and TRZ side effects in the CNS can be related to their main receptor-targeting mechanism, or (at least partially) could be explained by depletion of cholesterol, altered endocytosis and trafficking of receptors [42]. We propose that their potential effects on the body organs could be independent of their membrane protein binding, as the peripheral organs we investigated showed a uniform sterol biosynthesis inhibiting response. ARI is partial agonist of dopamine D2 receptors [43], while TRZ works through plethora of receptors and transporters (serotonin, histamine, adrenalin/noradrenalin) [44]. Nevertheless, many of these receptors are found at the periphery [45,46], and we cannot fully exclude the possibility that the sterol biosynthesis inhibiting effects arise by this mechanism.

Regardless of the exact mechanism of action, our data raise an important question: What are the possible clinical side effects of these medications in the periphery and what a clinicians can do to avoid them? To answer this question confidently, we first need to establish a catalogue of prescription medications with a 7-DHC elevating side effects [47,48,49], and try to avoid their simultaneous use [2,50,51], especially in patients with the *DHCR7*^+/-^ genotype [52] (who already have a baseline elevation of 7-DHC). However, based on the available data showing that concurrent use of 7-DHC elevating beta-blockers and psychotropic medications [53,54] can have adverse cardiac effects in patients [55], we suggest that clinical side effects could be related to heart muscle function [56]. Furthermore, the organ-specific side effects might depend on the exact combination of prescription polypharmacy, defined by the medications’ target organ to a large degree [55,57,58,59]. 

A uniform sterol biosynthesis inhibiting response to ARI and TRZ at the periphery also opens a question if the sharp rise in 7-DHC levels have ultimately unwanted effects on the functioning of all somatic organ systems. At the current time, we cannot answer this question with certainty, and this should be investigated in follow-up studies. However, psychiatric patients treated with multiple psychotropic medications often develop different metabolic disturbances [1,60,61], and we hypothesize that the 7-DHC elevation we observed might be a contributing mechanism to this unwanted clinical outcome. 

Potentially, pancreatic beta cell and glucose reabsorbing structures in the proximal tubules are the two most reactive oxygen species (ROS) sensitive tissues related to the development of metabolic syndrome, which can be damaged by 7-DHC [62,63]. Moreover, there are reports that link the oxidative stress pathway and cholesterol-induced apoptosis of beta cells [64], or proximal tubules acute stress response and dysregulation of cholesterol synthesis [65]. The possibility that ARI and TRZ act directly on cholesterol metabolism in macrophages, provoking a low-grade inflammation, and the development of metabolic syndrome should not be ignored [66]. Here, should be emphasized that the greatest increase in 7-DHC was found in the spleen. Due to the interconnectedness of glucose and cholesterol metabolisms [67], and cholesterol imbalance in all observed organs, systemic changes leading toward a vicious circle that ultimately results in metabolic syndrome cannot be excluded. As a result, we propose that in patients with chronic ARI and TRZ use the development of non-alcoholic liver steatosis, dyslipidemia, low-grade inflammation, muscle insulin resistance and changes in the stress response should be monitored.

We and others have previously documented the strong sterol biosynthesis inhibitory activity of ARI and TRZ on the developing and adult brain, neurons, and astrocytes [19,20,49,68] across various in vitro and in vivo models. Furthermore, it is apparent that the same 7-DHC elevating effects of TRZ can be also observed in human blood and postmortem brain [18,19,54,69]. Our current findings raise the possibility that the 7-DHC elevating effects of ARI and TRZ can impact the development of other tissues beyond the CNS. We know that the developmental disorder known as Smith-Lemli-Opitz syndrome (SLOS), a condition arising from two mutant copies of the DHCR7 gene, is characterized by malformations in multiple organ systems. In SLOS, the sharp elevation of 7-DHC (together with reduced cholesterol levels) disrupts systemic development of the offspring, potentially through interference with the sonic hedgehog (Shh) signaling [26,70,71]. In addition, recent human population studies suggest that DHCR7 inhibiting medications could be considered teratogens [72]. Thus, one should consider the potentially harmful effects of maternal ARI and TRZ utilization on the various organ systems of the offspring, not only the CNS. 

In most of the somatic organs and tissues examined we observed a summation or synergy on the 7-DHC elevating effects of ARI and TRZ polypharmacy. These findings, although observed in mouse tissues, are likely to translate to human physiology: the ARI-TRZ synergy/summation effect on DHCR7 inhibition is also seen in two human cell lines (HepG2 and human fibroblasts) and one mouse cell line (Neuro2a). In particular, peripheral human dermal fibroblasts represent a very powerful tool to investigate the translation potential of findings between mouse and human and has been successfully used to identify many critical physiological and pathophysiological processes in CNS, cardiovascular, and other research areas [73,74,75,76]. In addition, the translational value of our findings is further underscored by the conserved sterol biosynthesis pathway between the two species [77].

Our findings also raise multiple other mechanistic questions. First, we observed a synergistic effect of ARI-TRZ polypharmacy on 7-DHC in some tissues and in vitro systems, while noted a summative effect in others. The reasons for the differences behind the two responses remains unknown at the current time and can be related to differences in local sterol biosynthesis, clearance, antioxidant mechanisms, gene expression profile, epigenetic programming through ROS signaling, metabolic specificities, or other factors. Similarly, while DHCR7 inhibition should theoretically affect both **7**-DHD conversion to DES and 7-DHC conversion to CHOL (Bloch vs. Kandutsch–Russell pathway), the synergistic/summative effects are only observed in changes of 7-DHC amounts, and not DES levels.

## 5. Conclusions

In conclusion, both ARI and TRZ, either alone or in polypharmacy, strongly inhibit sterol biosynthesis and lead to elevation in 7-DHC levels and reduction in DES across all the somatic tissues we investigated. This is perhaps not achieved by their receptor binding action, but through inhibiting DHCR7 enzyme activity. This interference with sterol biosynthesis can potentially interfere with functioning or development of multiple organ systems, warranting further investigation.

## Figures and Tables

**Figure 1 biomolecules-12-01535-f001:**
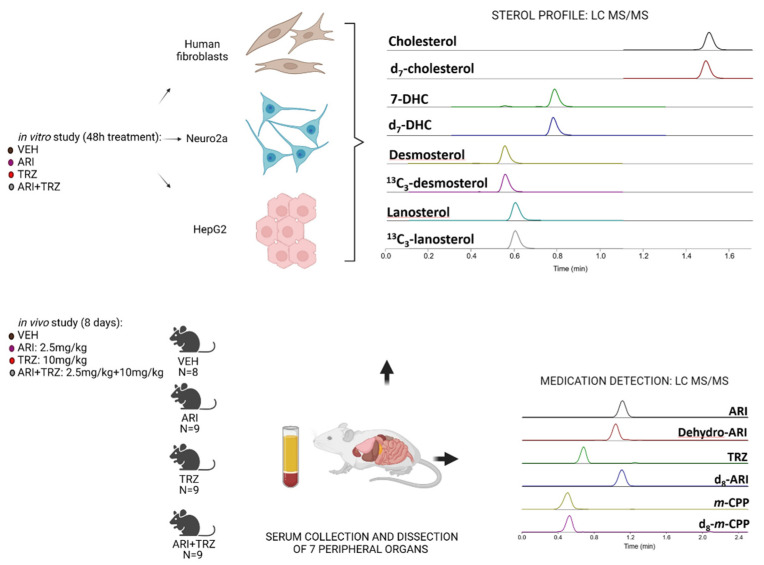
Experimental design of the study. In vitro experiments: HepG2, Neuro2a and human dermal fibroblasts were treated with ARI, TRZ or ARI+TRZ. CHOL, DES, 7-DHC, and LAN were analyzed with PTAD derivatization assay by LC-MS/MS. In vivo experiments: mice were treated with ARI (2.5 mg/kg), TRZ (10 mg/kg), ARI+TRZ (2.5 mg/kg + 10 mg/kg), or vehicle (VEH) for 8 days. Sera were collected and peripheral tissues were dissected (heart, lungs, pancreas, spleen, liver, kidneys, adrenal glands) for measurements of sterols, ARI, TRZ and their main metabolites by LC-MS/MS.

**Figure 2 biomolecules-12-01535-f002:**
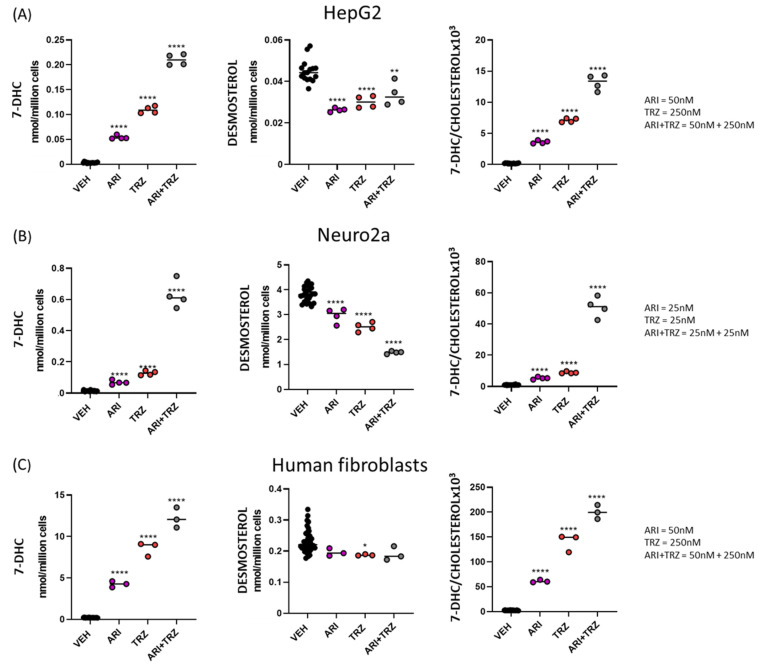
ARI, TRZ, and the combination of both significantly alter sterol biosynthesis. (**A**) HepG2, (**B**) Neuro2a, and (**C**) human fibroblasts. 7-DHC, DES levels and 7-DHC/CHOL ratio are shown. Data are presented for vehicle treatment, ARI, TRZ, and ARI + TRZ with concentrations indicated on the right side of each graph set. Treatments compared to vehicle exposure that reached significance are marked with black asterisks. * *p* < 0.05, ** *p* < 0.01, **** *p* < 0.0001.

**Figure 3 biomolecules-12-01535-f003:**
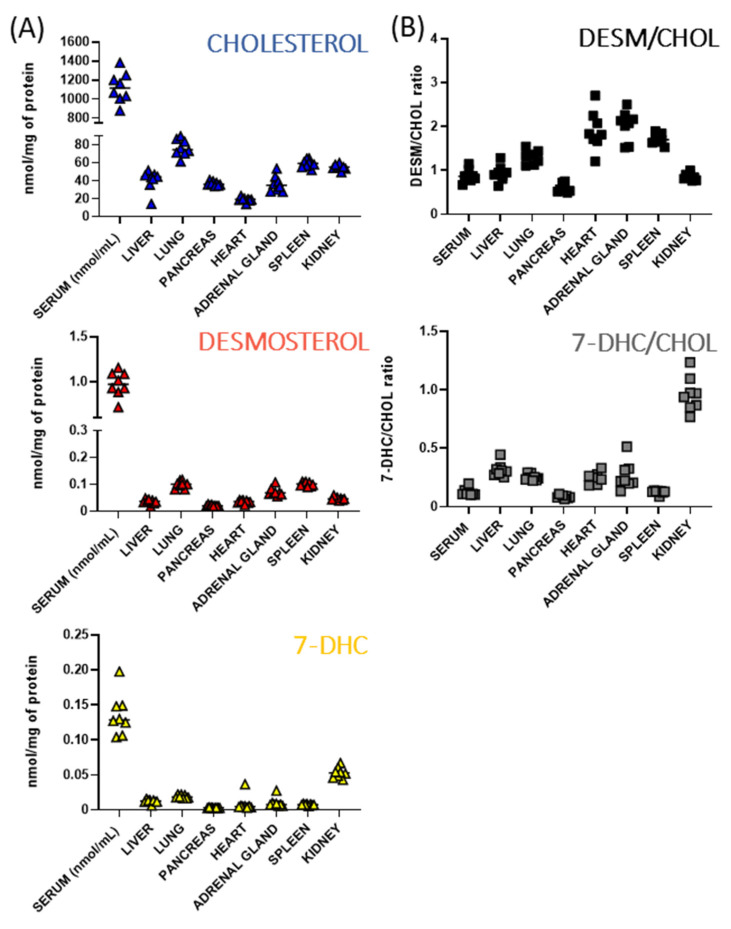
Sterol levels across serum samples and seven peripheral organs of adult mice. (**A**) Y-axis represents sterol levels (CHOL, DES, and 7-DHC), X axis represents sample types. Each symbol represents a single LC-MS/MS measured sample. (**B**) DES/CHOL and 7-DHC/CHOL ratios across the investigated organs and serum samples. Note the sterol content variability across the serum and examined organs.

**Figure 4 biomolecules-12-01535-f004:**
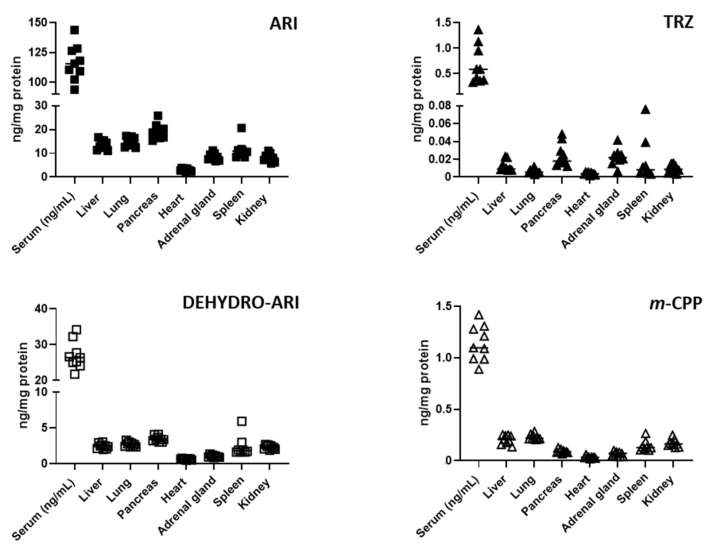
ARI and TRZ are detected in the serum and seven peripheral organs. Polypharmacy treatment ARI (2.5 mg/kg) + TRZ (10 mg/kg) data are reported. Symbols represents ARI, TRZ, or metabolite concentration in a single ARI + TRZ-treated sample. Both drugs and their metabolites were detected in serum and peripheral organs.

**Figure 5 biomolecules-12-01535-f005:**
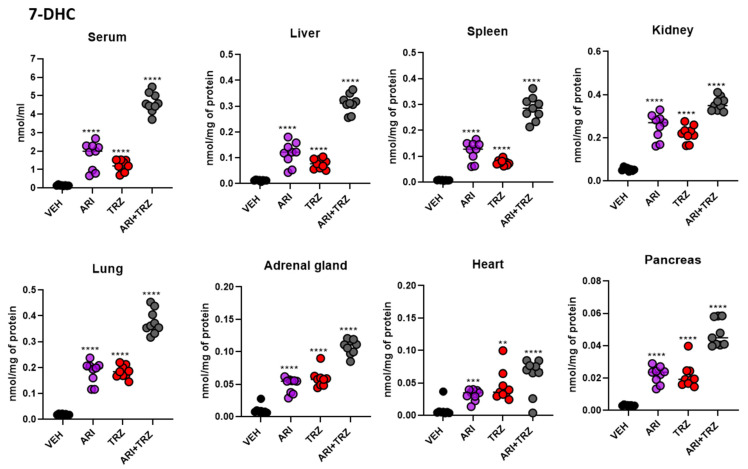
Combined ARI and TRZ treatment results in summative or synergistic increase in 7-DHC levels. X axis represents treated groups, Y axis denotes 7-DHC concentration normalized to nmol/mL (serum) or nmol/mg of protein (tissues). Black asterisk denotes significance compared to baseline levels. ** *p* < 0.01, *** *p* < 0.001, **** *p* < 0.0001. 7-DHC fold changes over vehicle treatment are presented in Appendix A, upper panel.

**Figure 6 biomolecules-12-01535-f006:**
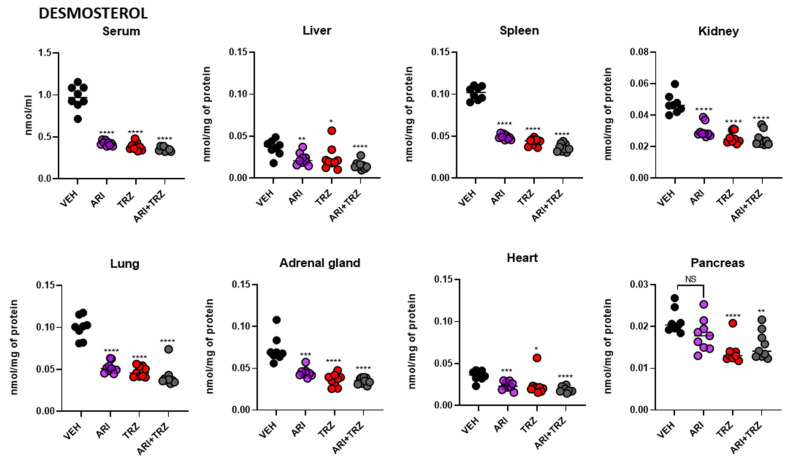
ARI and TRZ treatment effects on DES levels in the serum and seven peripheral organs of mice. Each symbol represents a sample from a single mouse. X axis represents treatment groups, Y axis represents DES concentration normalized to nmol/mL (in the case of serum) or nmol/mg of protein. Black asterisk denotes comparison to untreated baseline levels. Two-tailed groupwise *t*-test was used for statistical comparisons, * *p* < 0.05, ** *p* < 0.01, *** *p* < 0.001, **** *p* < 0.0001. Note the strong DES decrease by all treatments. Percentage of DES decrease for each sample type is presented in Appendix A, lower panel.

**Table 1 biomolecules-12-01535-t001:** Sterol levels in serum and peripheral organs of mice treated with ARI, TRZ, or ARI + TRZ. Serum levels are expressed as nmol/ml ± SEM, values for all organs are expressed as nmol/mg of protein ± SEM.

Sample	Treatment	7-DHC ± SEM	DES ± SEM
Serum (nmol/mL)	VEH	0.136 ± 0.011	0.975 ± 0.050
	ARI	1.754 ± 0.263	0.425 ± 0.010
	TRZ	1.194 ± 0.107	0.388 ± 0.016
	ARI + TRZ	4.616 ± 0.189	0.353 ± 0.008
Liver (nmol/mg)	VEH	0.012 ± 0.001	0.036 ± 0.003
	ARI	0.115 ± 0.015	0.023 ± 0.002
	TRZ	0.077 ± 0.006	0.024 ± 0.005
	ARI + TRZ	0.311 ± 0.012	0.015 ± 0.002
Spleen (nmol/mg)	VEH	0.007 ± 0.0004	0.101 ± 0.003
	ARI	0.120 ± 0.013	0.049 ± 0.001
	TRZ	0.076 ± 0.004	0.043 ± 0.001
	ARI + TRZ	0.285 ± 0.015	0.036 ± 0.002
Kidney (nmol/mg)	VEH	0.053 ± 0.003	0.047 ± 0.002
	ARI	0.252 ± 0.020	0.030 ± 0.002
	TRZ	0.219 ± 0.013	0.026 ± 0.001
	ARI + TRZ	0.357 ± 0.011	0.025 ± 0.002
Lung (nmol/mg)	VEH	0.019 ± 0.001	0.100 ± 0.005
	ARI	0.183 ± 0.014	0.052 ± 0.002
	TRZ	0.184 ± 0.008	0.047 ± 0.002
	ARI + TRZ	0.376 ± 0.015	0.041 ± 0.004
Adrenal gl. (nmol/mg)	VEH	0.010 ± 0.003	0.074 ± 0.006
	ARI	0.049 ± 0.004	0.045 ± 0.002
	TRZ	0.059 ± 0.004	0.036 ± 0.002
	ARI + TRZ	0.107 ± 0.004	0.035 ± 0.001
Heart (nmol/mg)	VEH	0.009 ± 0.0003	0.036 ± 0.001
	ARI	0.032 ± 0.003	0.024 ± 0.002
	TRZ	0.045 ± 0.008	0.024 ± 0.004
	ARI + TRZ	0.061 ± 0.009	0.019 ± 0.001
Pancreas (nmol/mg)	VEH	0.003 ± 0.0001	0.021 ± 0.001
	ARI	0.022 ± 0.002	0.018 ± 0.001
	TRZ	0.022 ± 0.003	0.014 ± 0.001
	ARI + TRZ	0.048 ± 0.003	0.016 ± 0.001

## Data Availability

The datasets used and/or analyzed during the current study are available from the corresponding author within reasonable limits.

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
