# Peer review of "Effects of Psychotropic Medication on Somatic Sterol Biosynthesis of Adult Mice"

_biomolecules, 2022, doi:10.3390/biom12101535_

Round 1

Reviewer 1 Report

In the present study, the Authors investigated the effects of Aripiprazole (ARI), Trazodone (TRZ) and ARI+TRZ polypharmacy on the post-lanosterol peripheral sterol biosynthesis in three cell lines (Neuro2a, HepG2 and human dermal fibroblasts) and an adult mouse model. Experimental design of the study was very well depicted in Figure 1 of the paper.

Overall, I believe and found this study very interesting, timely, well designed and scientifically sound: it adds something new to the existing literature on ARI and TRZ and especially on their combination. I have only some minor suggestions aimed to improve the high quality of the paper and these are outlined below.

1) I believe that in the Introduction a brief note on mechanisms of action of both aripiprazole and Trazodone world be beneficial to the reader with appropriate references (please see dois 10.9758/cpn.2021.19.4.780)

2) In thank discussion, thank Authors correctly wrote that ARI and TRZ side effects in the CNS can be related to their main receptor-targeting mechanism, or (at least partially) could be explained by depletion of cholesterol, altered endocytosis and trafficking of receptors. But which kind of side effects might be more involved by these drugs and their combination and what a clinician can do to avoid them?

3) As the Authors observed a synergistic effect of ARI-TRZ polypharmacy on 7-DHC in some tissues and in vitro systems, while noted a summative effect in others, do You recommend or not this combination in clinical practice as it’s commonly used (with TRZ at lower dosages to mainly improve sleep?

Author Response

  • I believe that in the Introduction a brief note on mechanisms of action of both aripiprazole and Trazodone world be beneficial to the reader with appropriate references (please see dois 10.9758/cpn.2021.19.4.780)

Added as suggested.

  • In thank discussion, thank Authors correctly wrote that ARI and TRZ side effects in the CNS can be related to their main receptor-targeting mechanism, or (at least partially) could be explained by depletion of cholesterol, altered endocytosis and trafficking of receptors. But which kind of side effects might be more involved by these drugs and their combination and what a clinician can do to avoid them?

We provide a brief discussion on this.

  • As the Authors observed a synergistic effect of ARI-TRZ polypharmacy on 7-DHC in some tissues and in vitro systems, while noted a summative effect in others, do You recommend or not this combination in clinical practice as it’s commonly used (with TRZ at lower dosages to mainly improve sleep?

We added a section on this very important point.

Reviewer 2 Report

I read with great interest the manuscript entitled Effects of psychotropic medication on somatic sterol 2 biosynthesis of adult mice by Balog et al.

In this work, the authors investigated the effect of the antipsychotic aripiprazole, the antidepressant trazodone and their combination on 3 cell lines and diverse tissues of a murine model. Their results showed that both drugs; either alone or in combination, inhibit sterol biosynthesis and lead to elevation in 7-DHC levels and reduction in DES across all analyzed tissues

They have reported a recent work in which they analyzed the same: individual and simultaneous treatment with aripiprazole and trazodone inhibiting sterol biosynthesis, but in the adult brain, using in vivo and in vitro experiments (Balog et al., 2022 doi: 10.1016/j.jlr.2022.100249). I think they should mention it in this manuscript in order to get the full story of the potential effect of these drugs and their combination on the brain.

There is also a previous work demonstrating  high 7DHC levels in postmortem brain of subjects who had taken trazodone (Cenik et al., 2022 doi: 0.1038/s41398-022-01903-3) that authors should mention in the discussion.

It is known that trazodone inhibits 7-dehydrocholesterol reductase and alter sterol concentrations in rodents, cell culture, human fibroblasts, and blood. Therefore, I think the main contribution of this study is to analyze the effects of these drugs in several tissues outside of nervous system.

M&M

-Please, specify the volume used in your aliquots (lines 117-121).

-how do you explain the results in HepG2 cells at dose 100nM of ARI in Fig S1?

Results

In lines 226-228, I think you refer to effect instead of affect “These data suggest that 226 under in vitro conditions the effects of ARI+TRZ polypharmacy are synergistic or sum- 227 mative, and preferentially effect 7-DHC levels.”

I guess something (a linker, or a word with an initial parenthesis) is missing in lines 375-377, please check the redaction of those lines:

“Also, we cannot exclude the possibility of metabolic cooperation between different cell types of the same tissue and equalization of the amount of cholesterol between cells by the mechanism of exocytosis) and liver (25-fold increase)”.

Discussion (line 456), please correct: Our study offers and alternative explanation…

Author Response

  • They have reported a recent work in which they analyzed the same: individual and simultaneous treatment with aripiprazole and trazodone inhibiting sterol biosynthesis, but in the adult brain, using in vivo and in vitro experiments (Balog et al., 2022 doi: 10.1016/j.jlr.2022.100249). I think they should mention it in this manuscript in order to get the full story of the potential effect of these drugs and their combination on the brain.

 Reference included as suggested.

  • There is also a previous work demonstrating  high 7DHC levels in postmortem brain of subjects who had taken trazodone (Cenik et al., 2022 doi: 0.1038/s41398-022-01903-3) that authors should mention in the discussion.

Reference included and discussed.

  • Please, specify the volume used in your aliquots (lines 117-121).

Volumes specified as suggested.

  • How do you explain the results in HepG2 cells at dose 100nM of ARI in Fig S1?

We suspect that it is some kind of technical error/user error. As we were unable to identify post hoc the source of the error, for full transparency we include the data.

  • In lines 226-228, I think you refer to effect instead of affect “These data suggest that under in vitro conditions the effects of ARI+TRZ polypharmacy are synergistic or summative, and preferentially effect7-DHC levels.”

Sentence fixed.

  • I guess something (a linker, or a word with an initial parenthesis) is missing in lines 375-377, please check the redaction of those lines: “Also, we cannot exclude the possibility of metabolic cooperation between different cell types of the same tissue and equalization of the amount of cholesterol between cells by the mechanism of exocytosis) and liver (25-fold increase)”.

Sentence fixed.

  • Discussion (line 456), please correct: Our study offers and alternative explanation…

Sentence fixed.